# Defining a Water-Soluble Formulation of Arachidonic Acid as a Novel Ferroptosis Inducer in Cancer Cells

**DOI:** 10.3390/biom14050555

**Published:** 2024-05-04

**Authors:** Zoe I. Day, Alyce J. Mayfosh, Amy A. Baxter, Scott A. Williams, Joanne M. Hildebrand, Thomas F. Rau, Ivan K. H. Poon, Mark D. Hulett

**Affiliations:** 1Department of Biochemistry and Chemistry, La Trobe Institute for Molecular Science, La Trobe University, Melbourne, VIC 3086, Australia; 21049418@students.latrobe.edu.au (Z.I.D.); alyce.mayfosh@wintermutebiomedical.com (A.J.M.); a.baxter@latrobe.edu.au (A.A.B.); i.poon@latrobe.edu.au (I.K.H.P.); 2Wintermute Biomedical, Geelong, VIC 3220, Australia; 3The Walter and Eliza Hall Institute of Medical Research, Parkville, VIC 3052, Australia; jhildebrand@wehi.edu.au

**Keywords:** ferroptosis, arachidonic acid, cell death, L-lysine, fatty acids, cancer

## Abstract

Here, we describe GS-9, a novel water-soluble fatty acid-based formulation comprising L-lysine and arachidonic acid, that we have shown to induce ferroptosis. GS-9 forms vesicle-like structures in solution and mediates lipid peroxidation, as evidenced by increased C11-BODIPY fluorescence and an accumulation of toxic malondialdehyde, a downstream product of lipid peroxidation. Ferroptosis inhibitors counteracted GS-9-induced cell death, whereas caspase 3 and 7 or MLKL knock-out cell lines are resistant to GS-9-induced cell death, eliminating other cell death processes such as apoptosis and necroptosis as the mechanism of action of GS-9. We also demonstrate that through their role of sequestering fatty acids, lipid droplets play a protective role against GS-9-induced ferroptosis, as inhibition of lipid droplet biogenesis enhanced GS-9 cytotoxicity. In addition, Fatty Acid Transport Protein 2 was implicated in GS-9 uptake. Overall, this study identifies and characterises the mechanism of GS-9 as a ferroptosis inducer. This formulation of arachidonic acid offers a novel tool for investigating and manipulating ferroptosis in various cellular and anti-cancer contexts.

## 1. Introduction

Arachidonic acid is a 20-carbon polyunsaturated fatty acid, characterized by the presence of four cis-double bonds. It is classified as an essential fatty acid belonging to the omega-6 family. In humans, the concentration of arachidonic acid in blood plasma is approximately 300 μM of total lipids [1,2]. The functions and metabolic pathways of arachidonic acid have previously been well described. Briefly, it serves as a crucial component of cell membranes, largely contributing to their fluidity. Additionally, arachidonic acid can be released from cell membranes through the Lands Cycle, enabling its participation in various cellular processes, including the synthesis of eicosanoids—a class of biologically active lipid mediators [1,3]. Notably, arachidonic acid has been implicated in many cell death pathways predominantly through alterations in membrane fluidity and the induction of oxidative stress [1].

Ferroptosis is a recently discovered form of regulated cell death initially described by Dixon et al. in 2012 [4]. This form of cell death is characterized by its dependence on iron and the accumulation of lipid reactive oxygen species or lipid peroxides, which serve as executioners of cell death. Mechanistically, these distinct features of ferroptosis differentiate it from other forms of cell death such as apoptosis, necrosis, and autophagy [5]. Current mechanisms of ferroptosis induction primarily revolve around perturbing the antioxidant capacity of cells, particularly focusing on the activity of Glutathione peroxidase 4 (GPX4). GPX4 plays a vital role in converting glutathione (GSH) to oxidized glutathione (GSSG), which possesses antioxidant properties that reduce lipid peroxides into their corresponding alcohols, thereby preventing the accumulation of lipid ROS within cells [6]. Well-known small-molecule inducers of ferroptosis, such as erastin, operate by inhibiting the activity of the system x_c_^-^ antiporter. This inhibition disrupts the exchange of extracellular cysteine with intracellular glutamate, impeding uptake and subsequently inhibiting GPX4 synthesis and its antioxidant function [7]. Additionally, other small-molecule inhibitors, including sorafenib, a kinase inhibitor, and sulfasalazine, a disease-modifying anti-rheumatic drug (DMARD), have been identified as inducers of ferroptosis. These compounds also directly or indirectly inhibit the activity of system x_c_^-^, thereby influencing downstream GPX4 and antioxidant activity [7,8].

Lipid metabolism, particularly the modulation of intracellular fatty acid composition, has been established as a crucial factor influencing ferroptosis sensitivity. The peroxidation of membrane fatty acids, such as arachidonic acid, has been linked to the induction of ferroptosis [9]. Therefore, due to the increased susceptibility of polyunsaturated fatty acids to intracellular oxidation processes, an elevation in intracellular levels of polyunsaturated fatty acids is recognized to drive ferroptosis [9,10]. As previously mentioned, arachidonic acid has been associated with cell death pathways [1]; however, more recent evidence has also implicated polyunsaturated fatty acids, like arachidonic acid, in the induction of ferroptosis [10,11,12].

In this study, we defined a novel fatty acid-based ferroptosis inducer called GS-9. Traditional applications of fatty acids have been limited by challenges related to solubility and the use of potentially toxic inorganic solvents for achieving solubility. To overcome these issues, GS-9 represents a new formulation comprising L-lysine and arachidonic acid. L-lysine facilitates the solubility of the fatty acid in solution. Through our investigations, we have demonstrated that GS-9 effectively induces cell permeabilization and subsequent death across various cancer cell lines. Importantly, this effect is not reliant on pro-apoptotic or pro-necroptotic proteins. Consistent with published studies on ferroptotic cell death, the utilization of C11-BODIPY, a lipid peroxidation marker, has revealed that GS-9 treatment promotes lipid peroxidation within cells. Furthermore, our findings indicate that GS-9-induced cell death can be counteracted by ferroptosis inhibitors such as ferrostatin-1, and that GS-9 causes an accumulation of Malondialdehyde (MDA), a toxic downstream metabolite of lipid peroxidation, further suggesting GS-9 induces ferroptosis. Additionally, we have identified that lipid droplets protect against GS-9-induced ferroptosis, as simultaneous inhibition of lipid droplet biogenesis enhances the GS-9 cytotoxicity. Finally, we have shown the involvement of Fatty Acid Transport Protein 2 (FATP2) in the uptake of GS-9, further uncovering its mechanism of action. Overall, GS-9 and the novel strategy for the solubilisation of fatty acids not only represents a unique tool for ferroptosis induction but also an exciting platform for further biological applications of fatty acids. 

## 2. Materials and Methods

### 2.1. Cell Culture 

The human lymphoma cell line U937 with caspase 3/7 knocked out as well as the Cas9 counterparts were kindly provided by Christine Hawkins (La Trobe University). The human colorectal carcinoma cell line HCT-116 with Bax and Bak knocked out as well as the Cas9 counterparts were kindly provided by Hamsa Puthalakath (La Trobe University). The human colorectal adenocarcinoma cell line HT-29 and lymphoma cell line U937 with Mixed-Lineage Kinase Domain Like Pseudokinase (MLKL) knocked out were kindly provided by Joanna Hildebrand (Walter and Eliza Hall Institute). Immortalised Bone Marrow-Derived Macrophages (iBMDM) from Gasdermin D (GSDMD)-knockout mice were kindly provided by James Vince (Walter and Eliza Hall Institute, Parkville Victoria, Australia). All other cell lines (wild type A549, Jurkat, U937, HT-29, HCT-116, HeLa) were purchased from ATCC unless otherwise stated. 

All media were supplemented with either 5% or 10% (*v*/*v*) FBS (Bovogen, Christchurch, New Zealand), 0.2% (*v*/*v*), MycoZap (Lonza, Basel, Switzerland), 100 U/mL penicillin and 100 U/mL streptomycin (Thermo Fisher Scientific, Waltham, MA, USA). HeLa, U937, and Jurkat cell lines were cultured in Roswell Park Memorial Institute Media (RPMI, Thermo Fisher Scientific). A549, HTC-116, HT-29, and iBMDM cell lines were cultured in Dulbecco’s modified Eagle’s medium (DMEM, Life Technologies, New York, NY, USA). All cell lines were cultured at 37 °C and 5% CO_2_.

### 2.2. Transmission Electron Microscopy (TEM)

Copper TEM grids featuring a formvar–carbon support film (GSCU300CC-50, ProSciTech, Kirwan, QLD, Australia) underwent a 60 s glow discharge in an Emitech k950x with a k350 attachment. Dilutions of 0.003 μM, 0.03 μM, 0.30 μM, and 3.00 μM of GS-9 were made up in water. Each grid received four-microliter drops of each GS-9 concentration, then allowed to adsorb for at least 30 s before subsequently being blotted with filter paper. Negative staining of the particles was performed using two drops of 2% uranyl acetate, with blotting after 10 s for each drop. The grids were then left to dry before the imaging process. Imaging was conducted using a Joel JEM-2100 transmission electron microscope equipped with an AMD NanoSprint II (Newspec, Myrtle Bank, SA, Australia), provided by JEOL Australasia (Brookvale, NSW, Australia).

### 2.3. Nanosight Tracking Analysis (NTA)

GS-9 was prepared at 3 μM in serum-free media for analysis on NS30031 (Malvern Panalytical, Malvern, UK), which was suitable for optimal particle concentration (determined as between 10–100 particles per frame, detection threshold level 6). Each sample was captured for three × 60 s measurements.

### 2.4. Dynamic Light Scattering (DLS)

GS-9 at 3 μM was added to 40 μL micro cuvette (Malvern Panalytical, Worcestershire, UK) and analysed using a Zetasizer Nano (Malvern Panalytical) at 25 °C.

### 2.5. Viability Assays 

Assays were performed as previously described except where indicated [13]. Briefly, all cell lines used in viability assays throughout this study were seeded into 96-well plates at 1 × 10^4^ cells per well in culture media. Adherent cell lines were seeded the day before to allow for adherence. All treatments were performed in triplicate. Wells containing culture media only were included as background control and cell only wells were included vehicle controls. All viability assays were performed over a 24 h period. For the relevant assays, arachidonic acid was solubilized at 55 mg/mL in Dimethyl Sulfoxide (DMSO) (Sigma-Aldrich, Burlington, MA, USA) before being diluted to appropriate concentrations. Some assays involved the use of inhibitors, including Ferrostain-1 (Sigma-Aldrich), Deferoxamine (Sigma-Aldrich), Vitamin E (α-Tocopherol) (Sigma-Aldrich), Q-VD-oPH (Abcam, Cambridge, MA, USA), IDN-6556 (Idun Pharmaceuticals, San Diego, CA, USA), Necrosulfonamide (NSA) (Merck #480073, Darmstadt, Germany), Necrostatin-1 (Sigma-Aldrich), Grassofermata (Sapphire Bioscience, Redfern, NSW, Australia) and Diglyceride acyltransferase inhibitor 1 (DGATi1; 2 mM) and Diglyceride acyltransferase inhibitor (DGATi2; 10 mM) (kind gifts from Professor Karla Helbig; collectively referred to as DGATi). Other viability assays involved the use of a BH3 mimetic apoptotic positive control which consisted of ABT-737 (Selleck Chemicals, Houston, TX, USA) and S63845 (Selleck Chemicals) (Jurkats 1.25 μM ABT-737 and 0.125 μM S63845, A549 ABT-737 at 5 μM and S63845 at 2 μM). Others used a necroptotic positive control designated TSI, comprising TNF-a (produced at Walter and Eliza Hall Institute [14]), Smac mimetic compound A (Tetralogic Pharmaceuticals, Malvern, PA, USA) and IDN-6556.

### 2.6. Confocal Microscopy 

For all imaging performed in this paper, cells were seeded in an 8-well chamber (Nunc, Roskilde, Sjælland, Denmark) at a density of 3 × 10^4^ cells/well in 1% BSA RPMI. If cells were adherent they were left overnight to adhere to wells. All imaging of live cells was performed at the La Trobe Bioimaging Platform on a Zeiss 800 Confocal Laser Scanning Microscope (Zeiss, Oberkochen, Germany) using 63× oil immersion at 37 °C and 5% CO_2_. The duration of imaging is noted in figure legends for respective experiments. Some assays involved the use of fluorescent dyes, including To-Pro-3-APC (Life Technologies), C11-BODIPY 581/591 lipid peroxidation sensor dye (2.5 μM) (Invitrogen, Waltham, MA, USA) and 4,4-Difluoro-1,3,5,7,8-Pentamethyl-4-Bora-3a,4a-Diaza-s-Indacene (BODIPY 493/503) (0.1 mg/mL) (Thermo Fisher Scientific). Images were processed by using software by Zen Image Analysis (Zen Blue Edition version 10.1.19043, Jena, Germany). 

### 2.7. Cell Death Measurement via Flow Cytometry

The Jurkat and A549 cell lines were plated in 96-well plates at densities of 2 × 10^4^ and 1.5 × 10^4^ cells/well, respectively, in culture media. A549 cells underwent an overnight incubation to facilitate adherence. Subsequently, cells were exposed to treatment with either GS-9 (300 μM) or a vehicle control and incubated for 3 h (Jurkats) or 6 h (A549s) at 37 °C with 5% CO2. Following incubation, cells were stained with Annexin-A5-FITC (AV) (BD Biosciences, San Jose, CA, USA) and To-Pro-3-APC (Life Technologies) in 1× AV Binding Buffer (BD Biosciences, Franklin Lakes, NJ, USA) and left on ice for 10 min before acquisition. The samples were analysed using a CytoFLEX flow cytometer (Beckman Coulter, Indianapolis, IN, USA) using CytExpert software version 2.5 (Beckman Coulter, Brea, CA, USA). Post-acquisition, data analysis was performed using FlowJo (BD Life Sciences, Franklin Lakes, NJ, USA) software version 10.8.

### 2.8. Lipid Peroxidation Flow Cytometry 

To investigate the ability of GS-9 to induce lipid peroxidation, the C11-BODIPY 581/591 lipid peroxidation sensor dye (Invitrogen, Waltham, MA, USA) was used. This dye is sensitive to oxidation such that upon peroxidation by ROS in cells it causes a shift in its fluorescence emission from red (∼590 nm) to green (∼510 nm). Jurkat and A549 cells were seeded at a density of 1.5 × 10^4^ cells/well and 2 × 10^4^ cells/well, respectively, in a 96-well plate in culture media. A549 cells were left to adhere overnight (37 °C and 5% CO_2_). Cells were then treated with GS-9 at concentrations listed in the figures with or without the presence of Ferrostatin-1 and incubated overnight. Before analysis, A549 cells were trypsinized and lifted before being stained with 2.5 μM C11-BODIPY 581/591 for half an hour. As described above, samples were analysed by flow cytometry.

### 2.9. MDA Accumulation

A549 cells (5 × 10^5^) were seeded in 6-well plates and incubated overnight. Cells were treated with Vitamin E, Ferrostatin-1, or Deferoxamine for 1 h before treating with 250 μM GS-9 for 16 h. Cells and lysate were collected, pelleted, washed in ice cold PBS and pellets snap frozen in nitrogen and stored at −80 °C until assay was performed. Levels of MDA were determined using a lipid peroxidation assay kit (Abcam, ab118970) according to the manufacturer’s instructions. Briefly, cell pellets were resuspended in MDA lysis buffer, sonicated and centrifuged at 13,000× *g* 10 min. MDA standards 2–20 μM were created. To each standard and lysate, TBA was added to generate an MDA-TBA adduct by heating at 95 °C for 1 h. Fluorescence of each standard and lysate was assessed in duplicate, excitation 532 nm, emission 553 nm.

### 2.10. Cell Titre-Glo Viability Assay 

A549 cells were seeded at 1 × 10^4^ cells/well in a 96 well plate before being treated with GS-9 (250 μM) in the presence of increasing concentrations of Vitamin E for 16 h before the CellTitre-Glo assay kit (Promega, Madison, WI, USA) was used to determine viability of treated cells. All values were corrected to a blank control and absorbance of the vehicle control was designated as 100% viability.

### 2.11. Western Blot Analysis of Caspase 3 and pMLKL

Adherent cell lines (A549 and HT-29) were seeded at a density of 5 × 10^5^ cells per well in a 6-well plate in culture media. Cells were incubated overnight to facilitate adhesion. Treatment conditions included GS-9 (at concentrations listed in figures), an untreated control (culture media) or positive controls. The positive control for the apoptosis assays was a BH3 mimetic (ABT-737 at 5 μM and S63845 at 2 μM). Positive controls for necroptosis assays were termed TSI, comprising TNF-a (produced at Walter and Eliza Hall Institute [14]), Smac mimetic compound A (Tetralogic Pharmaceuticals, Malvern, PA, USA) and IDN-6556. After treatment, cells were collected and lysed immediately using PMN lysis buffer (1% Nonidet P40, 10% glycerol, 1% TX-100, 3% NaCl 5M, 2% HEPES 1M) with protease inhibitor (cOmplete™, Mini, EDTA-free Protease Inhibitor Cocktail, Sigma). Protein concertation of cell lysate was determined using the Pierce Bicinchoninic acid (BCA) Protein Assay Kit (Thermo Fisher Scientific). NuPAGE LDS (lithium dodecyl sulphate) buffer (Invitrogen) and 1 M dithiothreitol were then added to lysates (enough for 20 µg protein. Lysates were then run on a 4–12% Bis-Tris gel (Thermo Fisher Scientific) in NuPAGE MES running buffer (Life Technologies, Carlsbad, CA, USA) using SeeBlue Plus 2 Protein Ladder (Invitrogen) at 120 V for 5 min, then 190 V for 35 min. Transfer was performed in transfer buffer at 20 V for 60 min onto a polyvinylidene fluoride (PVDF) membrane. Membranes were subsequently blocked and washed before being incubated with primary antibodies rabbit-anti-pro-caspase-3 (1:1000 in 1% BSA PBST) (Santa Cruz, Dallas, TX, USA, H-277 lot #12013) and rabbit-anti-phospho-MLKL (1:1000 in 1% BSA PBST) (Thermo Fisher Scientific, PA5105678) overnight at 4 °C. The following day, membranes incubated with secondary antibodies including goat anti-rabbit-HRP (1:5000, 0.1% PBTS with 5% skim milk powder) (Thermo Fisher) or sheep-anti-mouse HRP (1:5000, 0.1% PBTS with 5% skim milk powder) (GE Healthcare, Chicago, IL, USA) for 1 h at room temperature. Imaging was performed using the ECL prime detection reagent (Bio-strategy, Melbourne, VIC, Australia) and a Syngene gel documentation system (Syngene, Frederick, MD, USA). This process was then repeated with mouse-anti-ß-actin (1:1000 in 1% BSA PBST) (Sigma-Aldrich, cat# A2228, clone#AC-74) using sheep-anti-mouse HRP (1:5000, 0.1% PBTS with 5% skim milk powder) (GE Healthcare) secondary antibody.

### 2.12. BODIPY Uptake Assay 

Cells were seeded in 96-well plates, A549 cells at 2 × 10^4^ cells/well and Jurkat cells at 1.5 × 10^5^ cells/well in culture medium. A549 cells underwent an overnight incubation at 37 °C with 5% CO_2_ to facilitate adherence. To prevent cell disassembly during cell death, Cytochalasin-D (CytoD) (Sigma Aldrich), an inhibitor of actin polymerization, was employed. BODIPY 493/503 (0.1 mg/mL), Cytochalasin-D (5 mM) (Sigma Aldrich), and GS-9 (250 μM) were prepared in culture medium. GS-9 was introduced to cells at specified time points indicated in the figure legends. BODIPY 493/503 fluorescence was then determined using flow cytometry, as described above.

### 2.13. Involvement of Lipid Droplets in GS-9 Cytotoxicity Using DGAT Inhibition

A549 cells (2 × 10^4^) were seeded in a 96-well plate in culture media and incubated (37 °C and 5% CO_2_) overnight to facilitate adherence. The following were prepared in culture media prior to addition to cells: CytoD (5 μM), DGATi1 (2 μM), DGATi2 (10 μM) and GS-9 (250 μM). CytoD was added to all conditions/wells to prevent disassembly. A549 cells were treated with GS-9 with or without the addition of DGATi then incubated overnight. The following day cells were stained with BODIPY 493/503 (0.1 mg/mL) before analysis. Samples were analysed for BODIPY fluorescence by flow cytometry as described above.

### 2.14. Uptake Analysis with FATP2 Inhibitors

The day before completing the assay, A549 cells were seeded in a 96-well plate at 2 × 10^4^ cells per well in culture media and incubated (37 °C and 5% CO_2_) overnight to adhere. Treatments including BODIPY 493/503 (0.1 mg/mL), CytoD (5 μM) and Grassofermata (inhibitor of FATP2) (10, 25, 50 μM) were performed in complete media. Cells were initially treated with a CytoD/BODIPY 493/503 master mix for 5 min before the addition of Grassofermata and incubated for a further 5 min. An amount of 250 μM GS-9 was then added before a 4 h incubation. Samples were analysed for BODIPY fluorescence by flow cytometry, as described above.

### 2.15. Statistical Analysis

All statistical analysis was performed using GraphPad Prism software (GraphPad Software version 9.1.0). All experiments were performed 3 times unless otherwise stated in the corresponding figure legend. Data are presented as mean ± S.E.M., or as stated in the figure legends. Details of statistical analysis are included in corresponding figure captions. 

## 3. Results

### 3.1. Characterization of GS-9

In biological applications, fatty acids have traditionally required organic solvents or chemical modifications due to their insolubility in water. However, these alterations can compromise their activity and increase potential toxicities [15,16,17]. To address this, here we have introduced a novel strategy utilizing an amino acid to enhance fatty acid water solubility. In this study, we present a novel formulation called GS-9, a combination of arachidonic acid and L-lysine, where the addition of L-lysine renders arachidonic acid water soluble. Based on known chemical interactions and our previous work conducted with similar formulations [13], we hypothesize that the basic amine group of L-lysine deprotonates the carboxylic group of arachidonic acid, forming a stable ammonium carboxylate salt soluble in water (Figure 1A). 

We employed transmission electron microscopy (TEM) to further characterize GS-9. Our imaging revealed that GS-9 forms supramolecular vesicle-like structures in solution, with an elevated vesicle count corresponding to increasing GS-9 concentration (Figure 1B–E). Additional nanoparticle tracking analysis (NTA) and dynamic light scattering (DLS) confirmed consistent vesicle formation with an average vesicle diameter of around 100 nm (Figure 1F,G). We postulate that these vesicles are self-assembling and serve to shield hydrophobic arachidonic acid tails from aqueous environments by being enveloped in L-lysine. 

### 3.2. GS-9 Induces Cancer Cell Death

Arachidonic acid has previously been shown to have anti-cancer activity [10,18]. To investigate the anti-cancer activity of GS-9, we first examined its effects on a panel of human cancer cell lines. GS-9 dose-dependently inhibited the growth of A549, HeLa, HT-29, Jurkat, HCT-116 and U937 cells (Figure 2A). Similarly, arachidonic acid alone elicited a dose-dependent decrease in cell viability (Figure 2B), whereas L-lysine at 680 µM had negligible impact (Figure 2C), indicating that arachidonic acid is the active agent in GS-9. Interestingly, GS9 had a higher IC50 than arachidonic acid alone in Jurkat, HCT-116, HT-29 and U937 cell lines. A549 and Jurkat cell lines were used as model cell lines going forward. 

To confirm the anti-cancer activity of GS-9, flow cytometry was employed to assess cell death using a method by Jiang et al. (2016) [19]. The GS-9 treatment moderately increased Annexin V staining in A549 and Jurkat cell lines, indicative of phosphatidylserine exposure, and significantly increased To-Pro- in both cell lines, suggesting cell lysis (Figure 2D,E). Confirming this, live cell imaging of GS-9-treated cells showed increasing To-Pro-3 positivity in A549 and Jurkat cells along with observable morphological changes, suggesting some form of cell death was occurring (Figure 2F,G). 

### 3.3. GS-9 Likely Stimulates Lipid Peroxidation-Induced Ferroptosis in Cancer Cells

Ferroptosis is a recently characterized regulated cell death mechanism reliant on lipid peroxide accumulation [5]. Given arachidonic acid’s reactive C-C double bonds and related studies [10], we hypothesized that, like arachidonic acid, GS-9 may also induce lipid peroxidation. To investigate this, we employed C11-BODIPY 581/591, a fluorescent lipid peroxidation reporter dye that when oxidized causes a detectable shift in its fluorescent emission from red to green. Hence, we would expect to see an increase in green fluorescence if GS-9 is inducing lipid peroxide accumulation, being an indicator of ferroptosis. Using this dye, we observed a dose-dependent increase in FITC/green fluorescence upon GS-9 treatment in A549 cells via flow cytometry compared to vehicle controls, signifying elevated lipid peroxide levels (Figure 3A,B). Additionally, ferrostatin-1 (fer-1), a recognized ferroptosis inhibitor, was able to abolish this shift in green fluorescence (Figure 3A,B). The same results were also mimicked in Jurkat cells (Appendix A). In addition, live cell imaging of GS-9-treated A549 cells with the same C11-BODIPY 581/591 reporter dye reaffirmed these findings, showing increased green fluorescence in GS-9-treated cells compared to vehicle controls (Figure 3C), with Fer-1 again mitigating this shift, suggesting lipid peroxide accumulation in GS-9 treated cells, indicative of ferroptotic cell death (Figure 3C).

To further investigate ferroptosis induction by GS-9, we next measured levels of malondialdehyde (MDA) in GS-9-treated cells. MDA is a toxic secondary product from enzymatic/non-enzymatic lipid peroxidation [20] and a recognised marker of ferroptotic cell death. An MDA detection assay revealed GS-9-induced a dose-dependent MDA accumulation in A549 cells (Figure 3D) with similar results seen in Jurkat cells (Appendix A). We also investigated whether known ferroptosis inhibitors fer-1 (a synthetic antioxidant), deferoxamine (DFO) (an iron chelator) and Vitamin E (a natural antioxidant) [4,21,22] could protect against MDA accumulation in GS-9-treated cells. Fer-1 and DFO dose-dependently reduced MDA accumulation (Figure 3E,F) while Vitamin E (VitE) completely prevented MDA accumulation comparable to control levels (Figure 3G). This not only suggests GS-9 may be inducing ferroptosis, but also implicates MDA-related toxicities as a potential downstream mechanism of GS-9-induced cell death. To validate these findings, viability assays with these same inhibitors were also conducted, revealing significant protection against GS-9-induced cell death (Figure 3H–J). These findings propose GS-9 as a novel ferroptosis inducer.

### 3.4. GS-9 Is Not Pro-Apoptotic 

To establish GS-9 as a novel ferroptosis inducer, alternative forms of cell death were also investigated. In addition to ferroptosis induction, polyunsaturated fatty acids and their lipid peroxidation products have also been implicated in apoptosis [23,24]. Caspases 3 and 7 are key drivers of apoptosis induction [25]. Initially, caspase 3 activation following GS-9 treatment was probed through immunoblotting, revealing no cleavage of pro-caspase-3 compared to a BH3 mimetic pro-apoptotic control (ABT-737 and S63845 or A/S) that led to almost complete loss of pro-caspase 3 after 6 h (Figure 4A). To validate these observations, viability assays with pan-caspase inhibitors Q-VD-oPH and IDN-6556 also demonstrated that inhibition of caspase activity could not protect A549 cells from GS-9-induced cell death (Figure 4B,C), with consistent findings in Jurkat cells (Appendix A). Confirming the lack of caspase involvement in GS-9 toxicity, caspase 3/7 double knock-out cells [26] showed similar susceptibility to GS-9-induced cell death as wild-type controls, while the same knock out cells were significantly protected against the BH3 mimetic pro-apoptotic control (Figure 4D). Similar results were observed in Bax and Bak double knock-out cells [27], with Bax and Bak also being essential proteins in the apoptotic cell death pathway. Bax and Bak double-knock-out cells were equally as susceptible to GS-9-induced cell death to the wild-type controls (Figure 4E). These data demonstrate that GS-9 does not induce apoptosis and is not reliant on pro-apoptotic proteins for initiation of cell death, further implicating GS-9 as a novel ferroptosis inducer. 

### 3.5. GS-9 Does Not Induce Necroptosis

To rule out other forms of cell death and given the association of lipid peroxidation with lysosome membrane permeabilization (LMP) leading to necroptosis [28], we also aimed to rule out necroptosis in characterizing GS-9 as a novel ferroptosis inducer. We assessed mixed-lineage kinase domain-like protein (MLKL) and Receptor-Interacting Protein Kinase 3 (RIPK3) involvement, two proteins crucial for necroptosis induction [29]. These studies were performed in HT-29 and U937 cell lines as the model A549 and Jurkat cell lines used in the previous experiments are necroptosis-incompetent due to suppressed RIPK3 expression [30]. MLKL-knockout HT-29 cells exhibited similar susceptibility to GS-9 toxicity as wild-type cells, yet showed protection against a necroptosis-positive control (TSI) (Figure 5A). Additionally, immunoblotting demonstrated a lack of phosphorylated MLKL (pMLKL) in GS-9 treated HT-29 cells, but a presence of pMLKL in the necroptosis-positive control (Figure 5B). Furthermore, necrostatin-1 (Nec-1), an RIPK3 inhibitor, and Necrosulfonamide (NSA), an MLKL inhibitor, failed to protect against GS-9-induced cell death in HT-29 and U937 cells, while both remained effective against a positive necroptosis control (Figure 5C–F). These data suggest that necroptosis can be eliminated as a mechanism of GS-9-induced cell death. Moreover, pyroptosis was also ruled out as a mechanism, evidenced by Gasdermin D (GSDMD)-knockout cells lacking protection against GS-9-induced cell death (Appendix A). 

### 3.6. Lipid Droplets Play a Protective Role against GS-9-Induced Ferroptosis

To further understand how GS-9 may be inducing ferroptosis, we examined its localization. Fatty acids have previously been shown to localize to lipid droplets [10]. To determine whether GS9 would follow the same pattern, we employed BODIPY 493/503 staining, a fluorescent molecule that labels neutrally charged lipids. Time-course microscopy on A549 cells revealed increased BODIPY staining within lipid droplets 30 min post GS-9 treatment compared to controls which was maintained for 2 h (Figure 6A). These findings were also confirmed via flow cytometry analysis of GS-9-treated A549 cells in the presence of BODIPY 493/503. We again observed an increase in BODIPY fluorescence after GS-9 treatment, suggesting localization to lipid droplets (Figure 6B). Comparable BODIPY accumulation was also seen in Jurkat cells through imaging and flow cytometry (Appendix A). Previous studies have also suggested that lipid droplets can protect cells from fatty acid oxidation and oxidative cell death [10]. Given GS-9’s localization to lipid droplets, we hypothesized that these droplets might protect against GS-9-triggered cell death by potentially sequestering GS-9 and preventing lipid peroxidation. To test this, we used diglyceride acyltransferase 1 and 2 (DGAT1/2) inhibitors, collectively referred to as DGATi, which inhibit the biogenesis of new lipid droplets. Using BODIPY 493/503 as an indicator of intracellular GS-9, flow cytometry demonstrated reduced BODIPY fluorescence in DGATi and GS-9-treated cells compared to GS-9 only, indicating reduced GS-9 localization to lipid droplets upon DGATi treatment (Figure 6C,D). In support of our hypothesis, viability assays performed in the presence of DGATi increased GS-9’s potency and capacity to induce cell death (Figure 6E). Furthermore, imaging performed on A549 cells treated with GS-9 showed diminished BODIPY fluorescence in the presence of DGATi, and a concurrent increase in cell lysis as seen via an increase in To-Pro-3 staining (Figure 6F). 

In addition to its localization, we sought to understand how GS-9 was being taken up by cells. We explored the role of Fatty Acid Transport Protein 2 (FATP2), utilising the FATP2 inhibitor Grassofermata, which has previously been shown to be a non-competitive inhibitor that binds the AMP site of FATP2 [31]. While FATP2 is mainly expressed in liver and kidney cells, many cancer cells, including A549, display upregulated expression [32]. BODIPY 493/503-stained A549 cells treated with Grassofermata were subjected to flow cytometry to quantify BODIPY staining, as a representation of intracellular GS-9 levels. Intriguingly, FATP2 inhibition reduced BODIPY fluorescence, albeit not to control levels (Figure 6G,H), suggesting a role for FATP2 in GS-9 uptake but also suggesting alternate GS-9 uptake pathways may be involved. The potential for the involvement of other uptake pathways was supported by viability assays in A549 cells with Grassofermata, which failed to protect cells from GS-9-induced cell death (Figure 6I), indicating the potential involvement of other uptake mechanisms for GS-9.

## 4. Discussion

This study describes GS-9, a novel formulation of arachidonic acid and L-lysine that enhances the solubility of arachidonic acid. GS-9 forms vesicle-like structures in a solution and is capable of inducing cancer cell death via a pro-ferroptotic mechanism—a regulated cell death mechanism involving lipid peroxide accumulation. Cancer cells seem to protect themselves against GS-9-induced toxicity by sequestering GS-9 to lipid droplets, while the cellular uptake of GS-9 involves FATP2. The study ruled out apoptosis, necroptosis, and pyroptosis as mechanisms by which GS-9 induces cell death, confirming GS-9’s unique ferroptosis-inducing capability. This finding could have implications for cancer therapy and other biological applications of fatty acids upon further investigation and may enable the generation of a unique tool for studying ferroptosis. 

A common challenge in biological and therapeutic applications of fatty acids is their insolubility in water—a problem that has been addressed in this study through the generation of GS-9. Fatty acids often require organic solvents or chemical alterations for solubilization. Herein, we describe a novel strategy whereby amino acids like L-lysine are used to solubilise fatty acids such as arachidonic acid. Previous solubilization strategies include the use of lipophilic solvents [33,34], which limit the concentrations of fatty acids able to be used due to their inherently toxic nature. To overcome this limitation, other studies have attempted to make chemical modifications such as deuteration to increase the amount of metabolically available of arachidonic acid in biological settings [33]. At higher concentrations, these strategies often compromise activity and increase subsequent off-target toxicities of fatty acids in vivo [15,16,17]. Although the IC50 for GS-9 was higher in some tumour cell lines than the equivalent concentration of arachidonic acid alone, it should be noted that the solubilisation of arachidonic acid in these in vitro studies necessitated the use of the organic solvent DMSO. As amino acids are naturally occurring and non-toxic, the solubilization mechanism described herein presents a safer alternative, which offers the potential to enhance fatty acid utility at higher concentrations in in vivo settings and other biological contexts while mitigating safety concerns. Indeed, we have successfully used this approach previously to solubilise undecylenic acid using L-arginine^13^. That said, the investigation into the effect of l-lysine and other essential amino acids such as when consumed through the diet should be conducted to determine how levels may influence the effectiveness of GS-9 in vivo. 

Beyond its innovative role in enhancing fatty acid solubility, GS-9’s capacity to form supramolecular structures holds potential for drug delivery. Our findings, verified through DLS, NTA, and TEM, demonstrate GS-9’s ability to form vesicle-like structures in solution. These vesicle-like structures share morphological similarities with emerging nanotechnology-based drug delivery systems, such as nanoparticles and extracellular vesicles [35,36,37], suggesting the potential for the GS-9 encapsulation of diverse molecules. Notably, ongoing research explores nanomaterials for ferroptosis induction in cancer therapy [38,39]. Given GS-9’s analogous morphologies and ferroptosis-inducing properties, these formulations present a novel avenue for delivering small molecule anti-cancer agents. Further investigations might explore GS-9’s potential as a vehicle for delivering synergistic ferroptosis inducers like erastin or RSL3, thereby enhancing the anti-cancer efficacy.

Our initial hypothesis regarding GS-9’s involvement in the lipid peroxidation process considers its reactive C-C double bonds and its role in Eicosanoid synthesis [40]. Arachidonic acid is involved in the process of eicosanoid synthesis, where the C-C bonds in arachidonic acid undergo controlled lipid peroxidation to form a collection of bioactive lipids (eicosanoids). This is a form of enzymatic lipid peroxidation through cyclooxygenases, lipoxygenases, and cytochrome P450 enzymes [40]. Non-enzymatic lipid peroxidation also occurs within cells where the C-C bonds in arachidonic acid are susceptible to attach via cellular oxidants, which causes a cascade of Fenton chemistry and lipid peroxidation. Hence, given its propensity for enzymatic lipid peroxidation, the substantial influx of arachidonic acid in GS-9 is likely to overwhelm these pathways, leading to aberrant lipid peroxidation, thus driving ferroptosis. Interestingly, we observed differing sensitivities to GS-9 amongst the cancer cell line used in this study. This could be explained by cell intrinsic factors such as lipid composition, as dictated by ACSL4, which is known to dictate ferroptosis sensitivity [41]. Hence, further studies are need to fully elucidate how GS-9 effects both cellular sensitivity and induction of ferroptosis.

Ferroptosis, a recently defined form of regulated cell death, relies on iron and lipid peroxide accumulation for execution, yet much remains unknown regarding its regulation and induction. This calls for novel tools, such as ferroptosis inducers, for its study. Like arachidonic acid, GS-9 introduces a distinctive mechanism from canonical inducers like Erastin [7], Sulfasalazine [8], and RSL3 [42], which disrupt the GPX4 antioxidant system, reducing cellular antioxidant capacity through GSH production inhibition. This study broadens the scope of small-molecule ferroptosis inducers, suggesting that in addition to sensitising it to ferroptosis, arachidonic acid, in the form of GS-9, can alone induce aberrant lipid peroxidation and cell death. The use of GS-9 enables further insights into ferroptosis induction via lipid peroxidation rather than inhibition of cellular antioxidant capacity, enhancing the understanding of this cell death process. Further, being a novel and soluble formulation of arachidonic acid, GS-9 provides a better way to utilise fatty acids in biological systems and could potentially broaden the scope in which arachidonic acid and other fatty acids can be used.

Like previous studies on fatty acids [10], our findings suggest a protective role for lipid droplets against GS-9-triggered cell death. Should GS-9 serve as a ferroptosis inducer, our study proposes a mechanism to increase potency and reduce required concentrations for cell death induction through the concurrent use of DGAT inhibitors. Further to this, polyunsaturated fatty acids in general have been implicated as drivers of ferroptosis, as they are likely to undergo lipid peroxidation [43]. Thus, concurrent treatments with lipid droplet inhibitors, such as DGATi as employed here, could increase the effectiveness of various ferroptosis inducers, offering a strategy to enhance the anticancer potential of these agents.

Having systematically excluded apoptosis and necroptosis presently, GS-9 offers a platform for investigating ferroptosis in the context of aberrant lipid peroxidation, contributing to the understanding of this cell death mechanism. Subsequent research will delineate GS-9’s lipid peroxidation-induced ferroptosis pathway, solidifying its role as a novel ferroptosis inducer. Further, future research could leverage this groundwork to develop a platform to better utilise and solubilise fatty acids in biological settings. Some limitations of the study and future directions should also be noted. Although our initial experiments suggested that GS-9 induced strong cell death at 200 μM (with most cell lines tested having an IC50 of ~150 μM), we chose to routinely test a range of GS-9 concentrations (100, 200 and 300 μM) in subsequent studies to observe titration effects of GS-9 activity for experimental rigour. Hence, further studies could more closely investigate a titration between 150 and 200 μM to better explore indicators of ferroptosis at more relevant IC50 values across different cell lines. Furthermore, A549 and Jurkat were selected as representative cell lines for the apoptosis assays presented in this study as GS-9 had a ‘mid-range’ (129.1 μM) or ‘low’ (114.8 μM) IC50 against these cell lines, respectively, compared to the other cell lines (HeLa 124.1 μM, HCT-116 118.6 μM, HT-29 195 μM, U937 126.3 μM). It should also be noted that the different incubation periods in the assays performed for A549 and Jurkat were used to account for different kinetics of GS-9 activity against these cell lines. Therefore, studies beyond the initial characterisation of GS-9 could look more expansively at its effects across different cell lines in addition to the data presented for A549 and Jurkat cell lines.

## 5. Conclusions

In conclusion, this study introduces GS-9 as a novel ferroptosis inducer, addressing the challenge of fatty acid insolubility by using L-lysine to render arachidonic acid water-soluble. GS-9 holds promise as both a valuable tool for studying ferroptotic cell death and an innovative approach to exploring new anti-cancer therapies and a platform for the solubilisation of fatty acids.

## Figures and Tables

**Figure 1 biomolecules-14-00555-f001:**
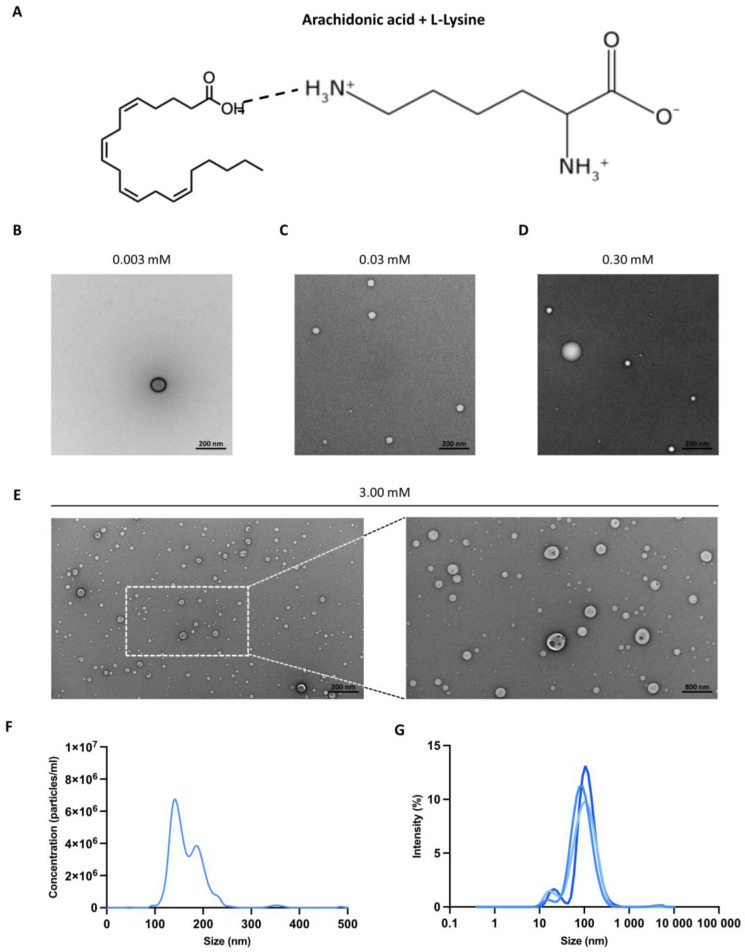
Biophysical properties of GS-9. (**A**) Schematic representation of the chemical structure and interactions of arachidonic acid and L-lysine in GS-9. TEM of GS-9 was performed at (**B**) 0.003 mM (25,000×), (**C**) 0.03 mM (30,000×), (**D**) 0.30 mM (25,000×), and (**E**) 3.00 mM (8000× and 25,000×). (**F**) Size of GS-9 supramolecular structures was determined via Nanosight Tracking Analysis after 24 h at 3 mM. (**G**) Supramolecular structure size was also evaluated via Dynamic Light Scattering immediately after dilution (0.03 mM) (n = 3 individual experiments shown).

**Figure 2 biomolecules-14-00555-f002:**
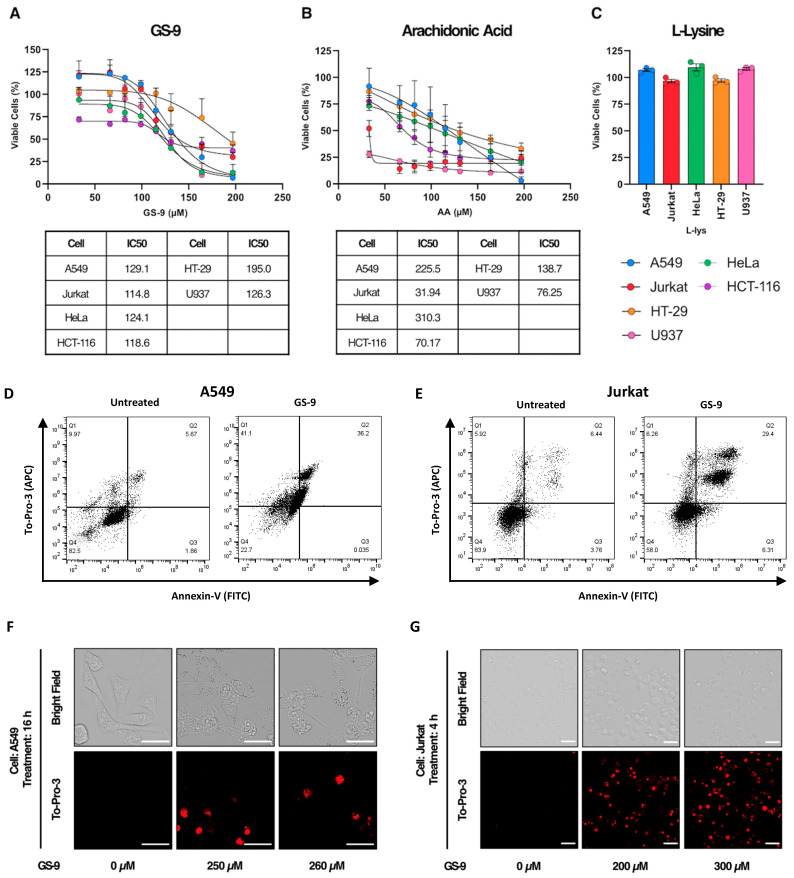
GS-9 inhibits cancer cell growth and causes cell lysis. (**A**) GS-9, (**B**) arachidonic acid or (**C**) L-Lysine (680 µM) were added to A549, Jurkat, HeLa, HCT-116, HT-29, and U937 cells for 24 h before MTT or MTS assays were performed to determine viability (viability calculated as a percentage of untreated controls n = 3, ±S.E.M.). IC50s (µM) for GS-9 (**B**) or arachidonic acid (**C**)-treated cell lines, are indicated. Both (**D**) A549 and (**E**) Jurkat cells were treated with or without GS-9 for 6 h and 3 h, respectively. Cells were stained with To-Pro-3-APC and Annexin V-FITC and analysed by flow cytometry. Both (**F**) A549 and (**G**) Jurkat cells were treated with GS-9 for 16 h and 4 h, respectively. Cells were then stained with To-Pro-3 and analysed via confocal time-course microscopy. Red fluorescence indicates To-Pro-3-positive cells (scale bar = 10 μm).

**Figure 3 biomolecules-14-00555-f003:**
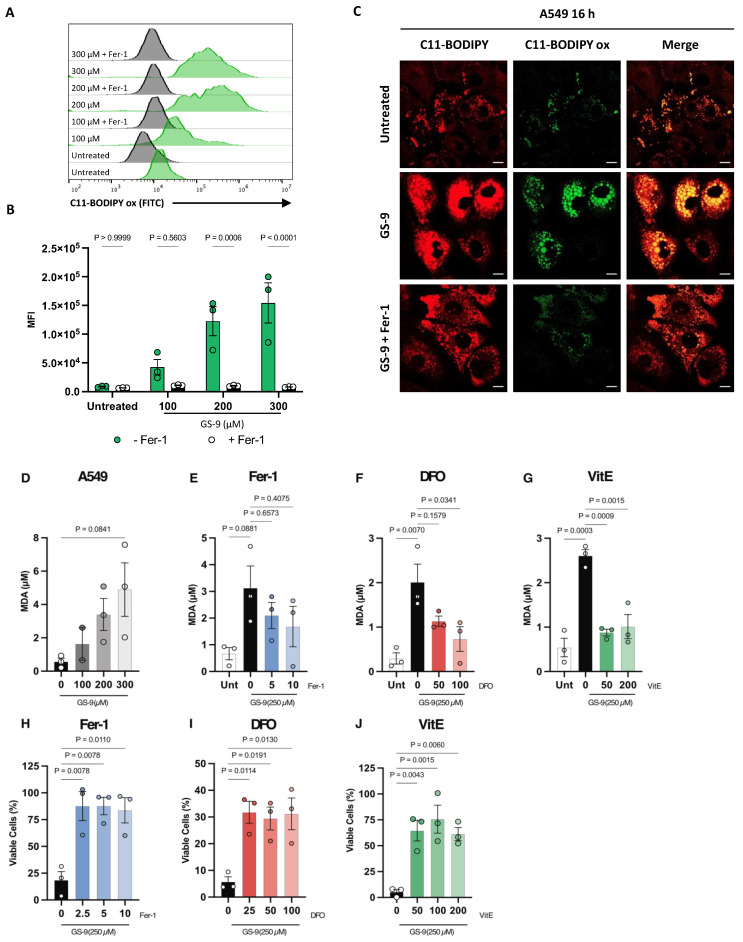
GS-9 likely induces ferroptosis in cancer cells. (**A**) GS-9 was added to A549 cells prior to being treated with C11-BODIPY 581/591 for 24 h. Histograms depicting oxidised C11-BODIPY 581/591 (FITC) levels were plotted after flow cytometry analysis. BODIPY C11 MFI was quantified in (**B**) (n = 3 ± S.E.M., two-way ANOVA, Šídák’s multiple comparisons test). (**C**) A549 cells were stained with C11-BODIPY 581/591and treated with GS-9 (250 µM) and imaged after 16h (scale bar = 10 μm). (**D**) A549 cells were treated with increasing concentrations of GS-9 for 24 h before MDA levels were analysed. A549 cells were treated with GS-9 (250 µM) in the presence of Fer-1 (**E**), DFO (**F**) and VitE (**G**) before MDA levels were analysed. A549 cells were treated with GS-9 (250 µM) in the presence of Fer-1 (**H**) or DFO (**I**) before undergoing MTT assay to determine viability. A549 cells were treated with GS-9 (250 µM) in the presence of VitE (**J**) before undergoing Cell Titre Glow assay to determine viability ((**D**–**J**): n = 2–3, ± S.E.M., one-way ANOVA, Tukey’s multiple comparison test).

**Figure 4 biomolecules-14-00555-f004:**
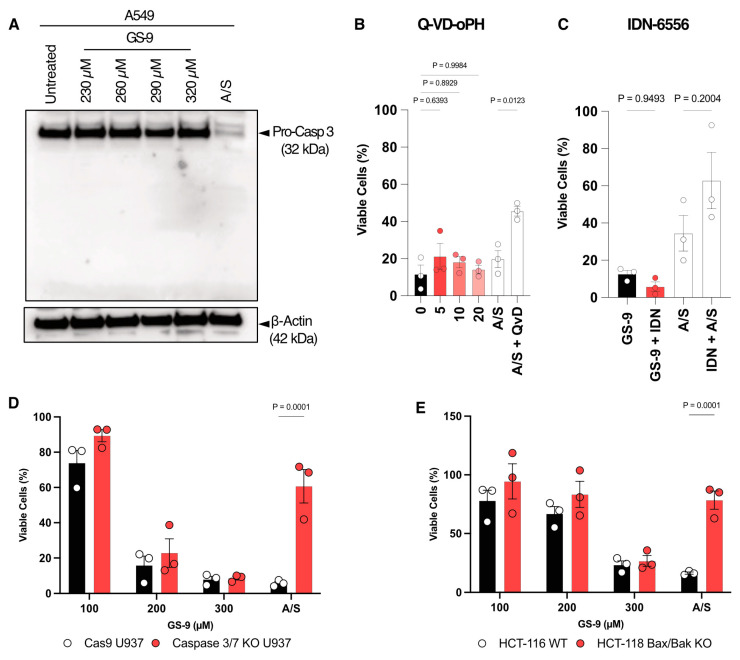
GS-9 is not reliant on pro-apoptotic proteins for the induction of cell death. (**A**) Increasing concentrations of GS-9 was added to A549 cells for 6 h in addition to an apoptosis-inducing positive control ABT-73 5, S63845 2 μM (A/S) for 6 h, before pro-caspase 3 levels were determined via immunoblotting. Apoptosis inhibitors (**B**) Q-VD-oPH (concentrations in figure) or (**C**) IDN-6556 (at 5 μM) were added to A549 cells prior to the addition of 250 μM GS-9 as well as a BH3 mimetic apoptosis positive control before MTT assays were performed to determine viability (viability calculated as a percentage of vehicle controls, n = 3, mean ± S.E.M., one-way ANOVA, Tukey’s multiple comparison test). (**D**) Increasing concentrations of GS-9 were added to caspase 3/7 DKO U937 cells and their WT counterparts before MTS assays were performed to determine viability (viability calculated as a percentage of vehicle controls, n = 3, ±S.E.M., two-way ANOVA, Šídák’s multiple comparisons test). (**E**) Increasing concentrations of GS-9 were added to Bax/Bak DKO HCT-116 cells and their WT counterparts before MTT assays were performed to determine viability (viability calculated as a percentage of vehicle controls, n = 3, ±S.E.M., two-way ANOVA, Šídák’s multiple comparisons test). Original Western Blots images can be found in the Appendix A.

**Figure 5 biomolecules-14-00555-f005:**
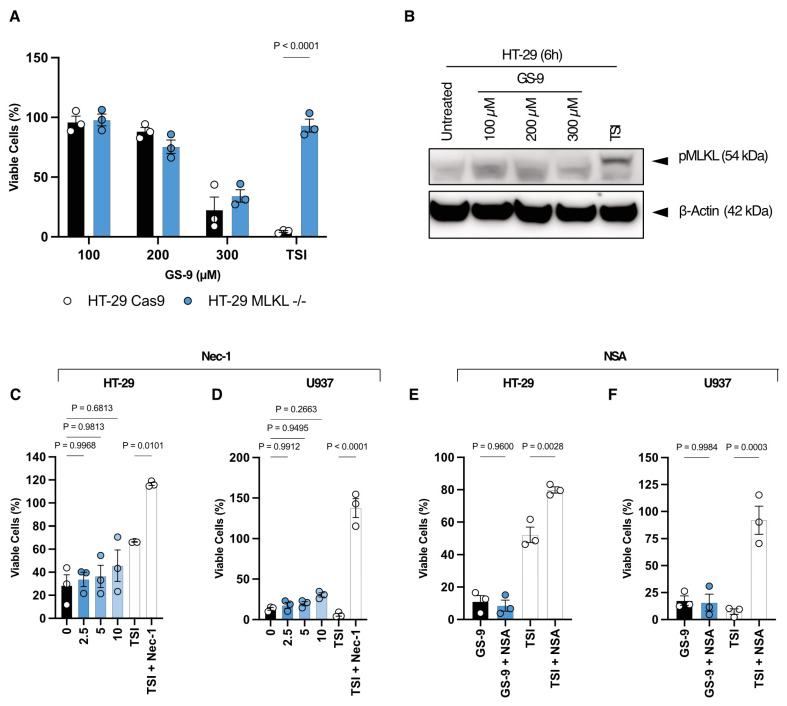
GS-9 does not induce necroptosis. (**A**) Increasing concentrations of GS-9 or a necroptosis-positive control (TSI) was added to MLKL−/− DKO HT-29 cells and their WT counterparts before MTT assays were performed to determine viability (viability calculated as a percentage of vehicle controls, n = 3, ±S.E.M., two-way ANOVA, Šídák’s multiple comparisons test). (**B**) Increasing concentrations of GS-9 were added to HT-29 cells for 6 h before levels of pMLKL were determined via immunoblotting. HT-29 (**C**) and U937 (**D**) cells were treated with a titration of Nec-1 in the presence of 250 μM GS-9 as well as a necroptotic-positive control before MTT or MTS assays were performed to determine viability (viability calculated as a percentage of vehicle controls, n = 3 ± S.E.M., one-way ANOVA, Tukey’s multiple comparison test). HT-29 (**E**) and U937 (**F**) cells were treated with NSA at 1 μM in the presence of 250 μM GS-9 as well as a necroptotic positive control before MTT or MTS assays were performed to determine viability (viability calculated as a percentage of vehicle controls, n = 3 ± S.E.M., one-way ANOVA, Tukey’s multiple comparison test). Original Western Blots images can be found in the Appendix A.

**Figure 6 biomolecules-14-00555-f006:**
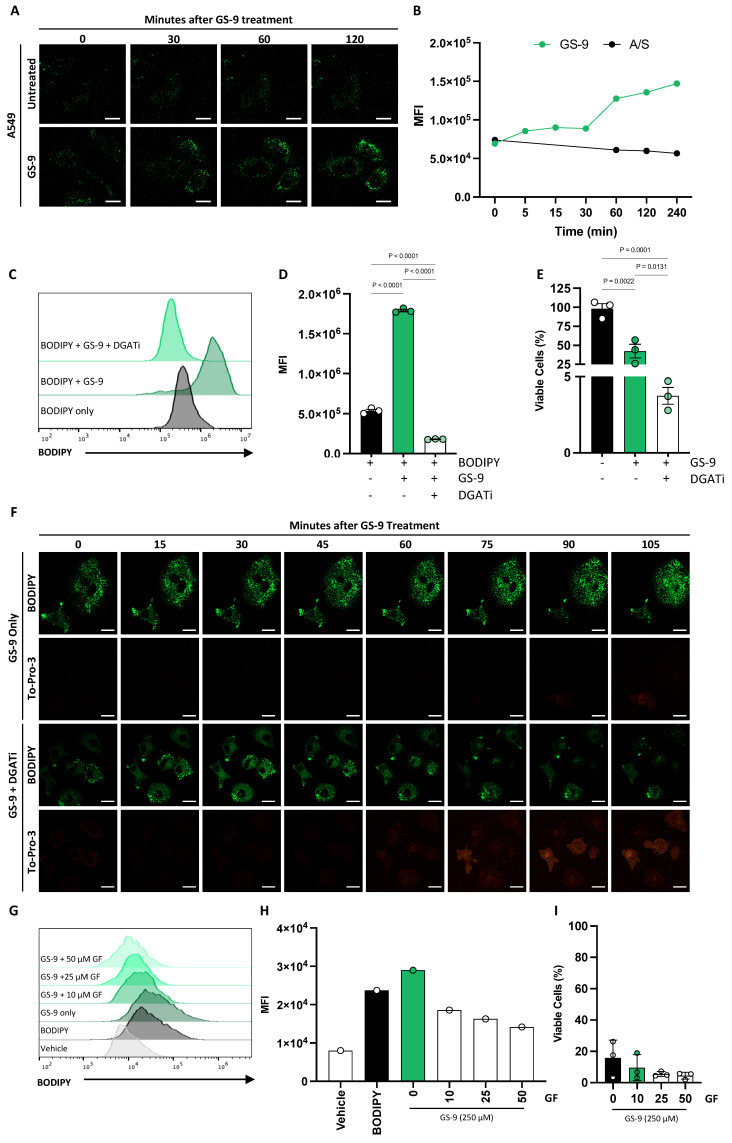
GS-9 localises intracellularly at lipid droplets. (**A**) GS-9 was added to A549 cells prior to being stained with BODIPY 493/503 before time course confocal microscopy was performed to determine GS-9 localisation (scale bar = 10 µm). (**B**) GS-9 was added to A549 cells prior to being stained with BODIPY 493/503 before GS-9 intracellular localisation at lipid droplets was analysed by flow cytometry, as indicated by BODIPY 493/503 MFI (representative of n = 3 experiments; replicates shown in Appendix A). GS-9 and DGATi were added to A549 cells for 4 h before staining with BODIPY. (**C**) Histograms depicting BODIPY fluorescence after flow cytometry analysis. BODIPY 493/503 MFI was then quantified in (**D**) (n = 3, ±S.E.M, one-way ANOVA, Tukey’s multiple comparison test). (**E**) GS-9 was added to A549 cells with or without DGATi for 24 h before MTT assays were performed to determine viability (viability calculated as a percentage of vehicle controls, n = 3, ±S.E.M., one-way ANOVA, Tukey’s multiple comparison test). (**F**) GS-9 was added to A549 cells prior to being stained with BODIPY 493/503 and To-Pro-3 with or without DGATi before time course confocal microscopy was performed (scale bar = 10 µm). (**G**) GS-9 in the presence of increasing concentrations of Grassofermata (GF) was added to A549 cells for 4 h before being stained with BODIPY 493/503. Histograms depicting BODIPY 493/503 fluorescence after flow cytometry analysis. BODIPY 493/503 MFI was then quantified in (**H**) (representative of n = 3 experiments; replicates shown in Appendix A). (**I**) GS-9 in the presence of increasing concentrations of Grassofermata (GF) was added to A549 cells for 4 h before MTT assays were performed to determine viability (viability calculated as a percentage of vehicle controls, n = 3, ±S.E.M.).

## Data Availability

Data sharing not applicable to this article as no datasets were generated or analysed during the current study.

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
