# Peer review of "Defining a Water-Soluble Formulation of Arachidonic Acid as a Novel Ferroptosis Inducer in Cancer Cells"

_biomolecules, 2024, doi:10.3390/biom14050555_

Round 1
Reviewer 1 Report
Comments and Suggestions for Authors
1. How were cellular iron levels affected if the culture media was not supplemented with iron? It is important since ferroptosis is induced by iron and ROS.
2. Arachidonic acid exhibited effects similar to GS-9 and even better effects in some cell types at the same concentrations (Figure 2, panels A and B). What was the reasoning behind using GS-9 nanoparticles to improve solubility?
3. GS-9 induced strong cell death effects at 200 uM (Figure 2A), but higher concentrations (>200 uM) were not studied initially. Subsequent experiments in Figures 2F, 2G, 3, 4, and 5 utilized higher GS-9 concentrations. What is the rationale for this inconsistency?
4. The rationale for choosing only A549 and Jurkat for the apoptosis assay (Figure 2 D-G) is unclear, especially since GS9 had a higher IC50 in all six cell lines. Why are A549 and Jurkat cells incubated for different periods in Figures 2D, 2E, 2F, and 2G?
5. The conclusions regarding the role of ferroptosis in mediating GS-9 effects are based on the effects of antioxidants (ferrostatin-1, tocopherol) and an iron chelator (DFO) on GS-9-induced cell death and lipid oxidation (measured by MDA and BODIPY). Oxidized phospholipids (PMID: 33684237), which were not analyzed in this study, are the main players that mediate ferroptotic signaling. While the study eliminates the roles of necroptosis and apoptosis, it does provide robust evidence that GS-9-induced cell death occurs primarily through ferroptosis.
Comments on the Quality of English LanguageAcceptable
Reviewer 2 Report
Comments and Suggestions for Authors
Day et al. introduce a novel ferroptosis inducer compound called GS-9 in their study. GS-9 addresses the solubility challenge by adding L-lysine moiety to enhance arachidonic acid water soluble. The authors present compelling data with appropriate control and elucidate the underlying mechanism to substantiate the efficacy of GS-9. There are some minor concerns.
The author highlights that GS-9 requires a higher dosage to achieve 50% cell death (higher IC50 concentration), suggesting that despite its enhanced water solubility, GS-9 needs a higher concentration for cellular uptake. Nevertheless, GS-9 still effectively induces significant cell death. The author did address this concern in lines 507-511. Perhaps these findings should be addressed more clearly in the Discussion (why the GS-9 yields less cytotoxicity?).
Line 320, the author states that “we hypothesized that, like arachidonic acid (ARA), GS-9 may also be susceptible to lipid peroxidation (LPO). “Susceptible” is confusing. "GS-9 may also induce LPO activity" would be a clearer and less ambiguous way to convey the idea of comparing ARA vs. GS-9.
Incorporating the Jurkat cell data in Figure 3E might be confusing. If the author intends to present Jurkat cell data, it would be beneficial to also include anti-LPO data. As Figures 3D, F-K focus on A549 cells, it may be helpful to move Figure 3E to the Supplemental section and display all treatment regimens for clarity.
WB of Casp3/7 and Bax/Bak proteins should be included to show that U937_KO and HCT-118_KO do not possess those pro-apoptosis genes. The author did provide ref 27 and 28 to support these cell lines, but WB should be included in this manuscript.
LPO and lipid droplet BODIPY probe are not clear and need to be clarified. BODIPY581/591 is for LPO and which BODIPY is used for lipid droplets (493/503?)
WB expression of FATP2 (+/- Grassofermata) should be included in Figure 6.
Reviewer 3 Report
Comments and Suggestions for Authors
In this article, entitled “Defining a water-soluble formulation of arachidonic acid as a novel ferroptosis inducer in cancer cells”, Day and colleagues describe a new compound (GS-9) as a novel water-soluble fatty acid-based formulation, potentially ferroptosis inducer. I found this work very
interesting, both for the potential therapeutic applications for the novelty in the synthesis of this formulation.
Although it is a promising work with new insights into the ferroptotic cell death mechanism, some points must be addressed to sustain author’s conclusion and for the manuscript to be published on Biomolecules.
1. First of all, the authors should demonstrate in their panel cells that GS-9 compound induces greater ferroptotic cell death than that observed with a well known ferroptotic-inducers (e.g ERA or RSL-3). To this aim, cell viability (FDA+/7-AAD-) and C11-BODIPY analysis are required to support the authors’s conclusions.
2. The choice of using some cell lines for some experiments to the exclusion of others is not very clear. It is advisable, where possible, to standardize the data obtained.
3. Is important that, during ferroptosis induction mediating by GS-9, the authors observe the increase expression levels of main ferroptotic markers ( such as SLC7A11, CHAC1, PTGS2) and reduction expression levels of GSH or GPX4 by qRT-PCR or western blotting analysis.
4. Levels of 4-HNE, as an other final lipid peroxidation product, must be addressed to MDA analysis. To this aim, the cell lines will be treated with GS-9 in presence or absence of Ferrostatin-1.
5. The authors use different tumor cell types with different response times to GS-9 treatment to induce cell death (Jurkat 3h and A549 6h). Does this mean that these cell lines are generally more reactive than each other? Additionally, they observe MDA accumulation after 16 hours, but generally lipid-ROS production is detected as cells begin to die. The authors can explain the different time used to detect cell death and peroxylipid production.
6. From biochemical point of view, is possible that the L-lysine in this compound interact specifically with MDA and 4-HNE and alters their structure, thus increasing the efficacy of GS-9 in tumor cells? Moreover, a new phospholipid such as Phosphatidylcholines (PCs) has been recent discovered, which has the highest potency and specificity for inducing ferroptosis compared with other.
7. Given that lysine is also an essential amino acid and therefore we consume it through diet, the authors can explain whether diet has an impact on the effectiveness of this compound in a potential therapeutic approach.
Round 2
Reviewer 1 Report
Comments and Suggestions for Authors
I am not satisfied with the authors' responses to my previous comments listed below. The authors should explain these weaknesses and correct the paper accordingly. Moreover, I recommend adding the "Limitations of the Study" section to describe these weaknesses.
My comments from the first round of review:
2. Arachidonic acid exhibited effects similar to GS-9 and even better effects in some cell types at the same concentrations (Figure 2, panels A and B). What was the reasoning behind using GS-9 nanoparticles to improve solubility?
3. GS-9 induced strong cell death effects at 200 uM (Figure 2A), but higher concentrations (>200 uM) were not studied initially. Subsequent experiments in Figures 2F, 2G, 3, 4, and 5 utilized higher GS-9 concentrations. What is the rationale for this inconsistency?
4. The rationale for choosing only A549 and Jurkat for the apoptosis assay (Figure 2 D-G) is unclear, especially since GS9 had a higher IC50 in all six cell lines. Why are A549 and Jurkat cells incubated for different periods in Figures 2D, 2E, 2F, and 2G?
